# Investigational Drugs for the Treatment of Postherpetic Neuralgia: Systematic Review of Randomized Controlled Trials

**DOI:** 10.3390/ijms241612987

**Published:** 2023-08-20

**Authors:** Miguel Á. Huerta, Miguel M. Garcia, Beliu García-Parra, Ancor Serrano-Afonso, Nancy Paniagua

**Affiliations:** 1Department of Pharmacology, University of Granada, 18016 Granada, Spain; huerta@ugr.es; 2Biosanitary Research Institute ibs.GRANADA, 18012 Granada, Spain; 3Area of Pharmacology, Nutrition and Bromatology, Department of Basic Health Sciences, Unidad Asociada I+D+i Instituto de Química Médica (IQM) CSIC-URJC, Universidad Rey Juan Carlos, 28922 Alcorcón, Spain; miguelangel.garcia@urjc.es; 4High Performance Experimental Pharmacology Research Group, Universidad Rey Juan Carlos (PHARMAKOM), 28922 Alcorcón, Spain; 5Clinical Neurophysiology Section—Neurology Service, Hospital Universitari de Bellvitge, Universitat de Barcelona-Health Campus, IDIBELL, 08907 L’Hospitalet de Llobregat, Spain; beliu.bellvitge@gmail.com; 6Department of Anesthesia, Reanimation and Pain Clinic, Hospital Universitari de Bellvitge, Universitat de Barcelona-Health Campus, IDIBELL, 08907 L’Hospitalet de Llobregat, Spain; a.serrano@bellvitgehospital.cat

**Keywords:** neuropathic pain, analgesic, first in class, AT2R, AAK1, LANCL, nerve growth factor, COX, opioid, NMDA

## Abstract

The pharmacological treatment of postherpetic neuralgia (PHN) is unsatisfactory, and there is a clinical need for new approaches. Several drugs under advanced clinical development are addressed in this review. A systematic literature search was conducted in three electronic databases (Medline, Web of Science, Scopus) and in the ClinicalTrials.gov register from 1 January 2016 to 1 June 2023 to identify Phase II, III and IV clinical trials evaluating drugs for the treatment of PHN. A total of 18 clinical trials were selected evaluating 15 molecules with pharmacological actions on nine different molecular targets: Angiotensin Type 2 Receptor (AT2R) antagonism (olodanrigan), Voltage-Gated Calcium Channel (VGCC) α2δ subunit inhibition (crisugabalin, mirogabalin and pregabalin), Voltage-Gated Sodium Channel (VGSC) blockade (funapide and lidocaine), Cyclooxygenase-1 (COX-1) inhibition (TRK-700), Adaptor-Associated Kinase 1 (AAK1) inhibition (LX9211), Lanthionine Synthetase C-Like Protein (LANCL) activation (LAT8881), N-Methyl-D-Aspartate (NMDA) receptor antagonism (esketamine), mu opioid receptor agonism (tramadol, oxycodone and hydromorphone) and Nerve Growth Factor (NGF) inhibition (fulranumab). In brief, there are several drugs in advanced clinical development for treating PHN with some of them reporting promising results. AT2R antagonism, AAK1 inhibition, LANCL activation and NGF inhibition are considered first-in-class analgesics. Hopefully, these trials will result in a better clinical management of PHN.

## 1. Introduction

Herpes zoster (HZ) is caused by reactivation of the varicella-zoster virus. Postherpetic neuralgia (PHN) is a complication of HZ that produces a chronic pain syndrome with a high incidence (5–30% of HZ patients) and increasing socioeconomic impact [1,2]. Patients develop severe pain persisting for more than three months after recovery from skin lesions [3]. PHN produces constant or intermittent pain in the absence of stimuli (spontaneous pain) with neuropathic pain (NP) characteristics, especially marked mechanical allodynia and thermal hyperalgesia [3,4]. The development of PHN involves mechanisms at both the central and peripheral nervous system levels [5,6], as summarized in Figure 1.

On the one hand, there is peripheral nociceptor sensitization with a reduction in the excitation threshold, the appearance of spontaneous ectopic discharges in peripheral and central axons, and a loss of descending pain inhibitory controls. On the other hand, central sensitization occurs through neuronal sensitization, along with spinal hyperexcitability [7].

Regarding treatment, clinical guidelines emphasize the importance of prevention and acute treatment after early diagnosis of HZ, all with the aim of avoiding the chronification of pain toward PHN [6,7,8]. This is crucial because once PHN is established, it is usually refractory to treatment with temporary or incomplete improvements despite multimodal therapy. First-line systemic drugs for PHN are tricyclic antidepressants (such as amitriptyline, nortriptyline, and desipramine) and gabapentinoids (gabapentin and pregabalin), despite producing adverse effects that limit their use [3,9]. If the patient does not respond to monotherapy, it is common to combine several drugs, usually opioids or topical treatments (e.g., lidocaine patches) [10]. However, the efficacy of pharmacological management remains suboptimal, so there is still a need for new treatments [1,11]. The aim of this systematic review is to summarize the results of drugs evaluated in Phase II/III/IV clinical trials between 2016 and 2023 in order to provide an overview and make predictions for the molecules and pharmacological targets that may be available in the future therapeutic arsenal. The chosen time period specifically corresponds to the year preceding the approval of Shingrix (shingles vaccine) by the FDA.

## 2. Materials and Methods

### 2.1. Protocol and Registration

The methodology used in this review was specified in advance and documented in a protocol that was registered in the CRD (Centre for Reviews and Dissemination) York website PROSPERO (International Prospective Register of Systematic Reviews) under the registration ID CRD42023423305. The study was performed adhering to the last version (2020) of PRISMA (Preferred Reporting Items for Systematic Reviews and Meta-Analyses) guidelines on systematic reviews and meta-analyses [12].

### 2.2. Review Question Statement and PICOS Elements

What drugs have been evaluated in Phase II/III/IV clinical trials for the treatment of PHN since 2016 (after Shingrix vaccine approval by the FDA)?

(P) Adult patients with PHN;

(I) All drugs being evaluated in Phase II/III/VI clinical trials for PHN;

(C) Control group;

(O) Pain reduction, quality of life improvement and side effects; and

(S) Randomized controlled trials.

### 2.3. Information Sources and Search Strategy

A comprehensive systematic search was performed in three databases: Medline, Web of Science, Scopus and in the register ClinicalTrials.gov from date 1 January 2016 to 1 June 2023without restriction in language. These three databases were chosen because they are the most used for biomedical purposes. The search strategy per database included the following:

Medline: “postherpetic neuralgia” OR “herpetic neuralgia” OR “herpetic neuropathy” OR “postherpetic neuropathy” OR “Neuralgia, Postherpetic” [Mesh] filter: Randomized Controlled Trials and time limitation (2016–2023)

Web of Science: (“postherpetic neuralgia” OR “herpetic neuralgia” OR “herpetic neuropathy” OR “postherpetic neuropathy”) AND (“randomized controlled trial” OR RCT OR “randomized controlled trial”) (All Fields) and 2016 or 2017 or 2018 or 2019 or 2020 or 2021 or 2022 or 2023 (Publication Years)

Scopus: TITLE-ABS-KEY (“postherpetic neuralgia” OR “herpetic neuralgia” OR “herpetic neuropathy” OR “postherpetic neuropathy” AND “randomized controlled trial” OR RCT OR “randomized controlled trial”) AND (LIMIT-TO (PUBYEAR, 2023) OR LIMIT-TO (PUBYEAR, 2022) OR LIMIT-TO (PUBYEAR, 2021) OR LIMIT-TO (PUBYEAR, 2020) OR LIMIT-TO (PUBYEAR, 2019) OR LIMIT-TO (PUBYEAR, 2018) OR LIMIT-TO (PUBYEAR, 2017) OR LIMIT-TO (PUBYEAR, 2016))

### 2.4. Inclusion and Exclusion Criteria 

Inclusion Criteria: randomized controlled trials in which adult patients with PHN were treated with any drug trialed in Phase II, III or IV studies.

Exclusion Criteria: review articles, systematic reviews, in vitro experiments, animal studies, studies including no relevant information and violation of any of the above inclusion criteria.

### 2.5. Article Selection 

Titles and abstracts of studies were retrieved using the search strategy by two review authors (MAH and BGP) in a blind manner to identify studies that potentially met the inclusion criteria. Full texts of these potentially eligible studies were retrieved and independently assessed for eligibility by two team members (MAH and BGP). The selection process was completed using the software Rayyan (Rayyan Systems Inc., Cambridge, MA, USA). Disagreements between them over the eligibility was resolved through discussion with a third reviewer (ASA).

### 2.6. Data Extraction 

Extracted information included drug evaluated, pharmacological target, administration route, dosage, clinical trial code, phase, and completion date. Two authors (MAH and BGP) extracted data independently (blind). Discrepancies were identified and resolved through discussion with a third author where necessary (ASA).

## 3. Results and Discussion

### 3.1. Article Selection

A summary flow chart is shown in Figure 2. The search yielded 558 articles (418 articles after removing duplicates). Then, the titles and abstracts were evaluated, and 344 articles were excluded for the following reasons: 150 because they were not original articles (mainly reviews or systematic reviews), 87 due to the population being different to PHN patients (mainly patients with other forms of neuropathic pain), 103 because they did not use a pharmacological intervention (mainly cognitive or physical therapies), and 4 for the reason that they evaluated an outcome different to pain (mainly evaluating pharmacokinetics). Of the 74 remaining full-text articles that were assessed for eligibility, 56 were excluded for different reasons: 10 articles because they were reviews (mainly meta-analyzing data of published RCT), 6 articles because the population was not PHN patients exclusively, 8 articles because the drug was used before PHN was established (for prevention), 30 studies because they were not Phase II, III or IV RCT, and 2 articles due to the intervention were procedures instead of drugs. Finally, for this systematic review, we selected a total of 18 studies that evaluated in Phase II, III or IV clinical trials the efficacy of different molecules to alleviate established PHN.

### 3.2. Study Characteristics

The complete list of included articles with detailed characteristics (pharmacological target, drug, route, dosage, clinical trial code, phase and completion date) is shown in Table 1. The most repeated main pharmacological target, referring to the one that explains the analgesic effect, was subunit α2δ of voltage-gated calcium channels (VGCCs) [13,14,15,16,17,18], which was followed by Voltage-Gated Sodium Channels (VGSCs) in its different subtypes [19,20] and mu opioid receptor [21,22,23]. Other mechanisms included were Angiotensin II Type 2 Receptor (AT2R) [24], Cyclooxygenase-1 (COX-1) [25], Adaptor-Associated Kinase 1 (AAK1) [26], Lanthionine Synthetase Component C-like protein (LANCL) [27], an unknown mechanism [28], N-Methyl-D-Aspartate Receptor (NMDAR) [29] and Nerve Growth Factor (NGF) [30]. The majority of the included clinical trials (14 of 18) were registered in the registry of the United States National Library of Medicine (NLM) at the National Institutes of Health (clinicaltrials.gov). The rest were registered in Chinese (ChiCTR), Japanese (JRCT) and Australian (ANZCTR) registries, with 2, 1 and 1 trials registered, respectively. Eight of them were Phase III trials, seven were Phase II and the remaining 3 were Phase IV trials. The most common administration route was oral, but topical, intravenous, intranasal, transdermal and subcutaneous were also used. The most important efficacy information on primary outcomes is summarized in Table 2. Information about other secondary outcomes can be found in Appendix A. Safety main results are summarized in Table 3, while information about serious adverse events can be found in Appendix A.

### 3.3. Future Insights According to Pharmacological Target

#### 3.3.1. Angiotensin II Type 2 Receptor Antagonism

There is increasing evidence implicating the Renin–Angiotensin System (RAS) in multiple facets of NP [32]. Specifically, there are many studies highlighting the location of type 2 receptors in important pathways for NP signaling, and there are numerous preclinical studies showing efficacy in animal models of NP [33]. However, there is also controversy regarding the role of Angiotensin II Type 2 receptors in pain, with studies suggesting both pro-algesic and anti-hyperalgesic characteristics [34]. Olodanrigan (EMA401) is a highly selective Angiotensin II Type 2 receptor antagonist developed by Novartis [35]. It has been evaluated in Phase II in the EMPHENE study [24] and in ACTRN12611000822987 [36] for PHN. It has also been evaluated for diabetic peripheral neuropathy (DPN) in EMPADINE [24]. In all studies, it showed promising results in both efficacy and safety (absence of drug-related SAEs). However, the last two trials were prematurely halted due to observed long-term hepatotoxicity in preclinical studies, although it was not observed in these clinical trials [24]. CFTX-1554 is the alternative Angiotensin II Type 2 antagonism proposed by Confo Therapeutics [37]. Although it is in an earlier phase of clinical development, it has recently established a collaboration with Eli Lilly to continue its development in Phase II [38].

#### 3.3.2. Voltage-Gated Calcium Channel (VGCC) α2δ Subunit Inhibition

The involvement of this target in NP has been known for years [39]. Inhibitors such as gabapentin or pregabalin (gabapentinoids) have been used for its treatment for years [40], despite their initial indication being for the treatment of epilepsy [41,42]. Pregabalin, as mentioned, is a first-line treatment for PHN, and its superiority over gabapentin is demonstrated [43]. New dosage forms of pregabalin have been developed for improving adherence to treatment, such as controlled-release pregabalin [16]. There is evidence suggesting that using strategies that involve minor dosing frequency of oral medications results in better adherence to treatment [44], which is confirmed by a meta-analysis comparing adherence of different dosages strategies [45,46]. Another new dosage form (once-daily sustained-release pregabalin formulation, YHD1119 tablets [18]) was developed in order to solve some problems related with variable absorption (attributed to the narrow absorption window of pregabalin [47]) of a previous approved controlled-release formulation (LYRICA^®^ CR, pregabalin extended-release tablets). YHD1119 uses a technology based on a floating and swelling gastroretentive drug delivery system [48]. In addition, the use of pregabalin is usually accompanied by undesirable adverse effects, such as dizziness, drowsiness, and peripheral edema. Because of this, in recent years, researchers have developed molecules also capable of blocking this target and being effective but with less central adverse effects. One example is mirogabalin, which has a high potency and selectivity to the α2δ-1 subunit [49,50]. Furthermore, in vitro studies confirmed that it has a slower dissociation rate from α2δ-1 than α2δ-2 compared with pregabalin [51]. Mirogabalin showed good results in several clinical trials including Phase III trial NEUCOURSE (NCT02318719) for PHN [52], which was complemented by an open label extension for long-term assessment [15]. Finally, it was approved in Japan for treating PHN in 2019 (Tarlige^®^) [53], but it is not approved yet by the EMA or FDA. Another example is crisugabalin (HSK16149), which was developed by the Chinese pharmaceutical company Haisco Pharmaceuticals. It has greater selectivity than pregabalin for the receptor, which could improve short- and long-term analgesia as well as reduce its central adverse effects. A Phase III trial in PHN patients was recently completed, but data have not been published (NCT05140863) [13]. In addition, a large Phase II trial with 300 patients (NCT05763550) [54] has recently begun, and the results are expected in July 2023. Recently, positive results were published of a Phase III evaluating HSK16149 for DPN [55]. Moreover, preclinical data demonstrate greater efficacy and fewer central adverse effects compared to pregabalin [56]. The most commonly reported AEs for HSK16149, mirogabalin, pregabalin CR, sustained release pregabalin, and standard pregabalin when compared with placebo included somnolence, dizziness, weight increase and edema. SAEs occurred more frequently in all mirogabalin groups at higher doses; incidence was significantly higher with pregabalin CR and standard pregabalin compared to placebo.

#### 3.3.3. Voltage-Gated Sodium Channel (VGSC) Blockade

The importance of Voltage-Gated Sodium Channel (VGSC) blockade in pain transduction is well known. A clear probe is that loss-of-function mutations in Nav1.7, a VGSC subtype, causes congenital insensitivity to pain in humans [57]. There are eight other α subunit isoforms (Nav1.1–1.9) which also are involved in pain neurotransmission [58]. Lidocaine, which is a classical drug widely used as a local anesthetic, was approved by the FDA for PHN in 1999 as topical administration (Lidoderm^®^ patch) [59]. Lidocaine is also FDA approved and widely used for local or regional anesthesia by infiltration techniques (nerve blocks). However, although there are some clinical experiences of systemic treatment of PHN with lidocaine [60,61], it is not approved for PHN. Recently, a Phase III clinical trial was performed evaluating the intravenous infusion of lidocaine for PHN, and the results showed an improvement of pain management, less analgesic consumption and faster recovery [20]. The most common AEs after intravenous lidocaine infusion were somnolence, dry mouth, and mild peripheral numbness. Interestingly, there are several novel VGSC blockers in clinical development for the treatment of NP, and some of them have been trialed for PHN [57]. One example is eslicarbazepine (BIA 2-093), which was evaluated in 2012 for PHN and DPN (NCT01124097 [62], NCT01129960 [63]). These studies were terminated early due to a high incidence of adverse events (e.g., vertigo, nausea, fatigue, dizziness, and headaches). A more recent compound included in this systematic review is funapide, a topical selective Nav1.7 and Nav1.8 VGSC blocker [64], which was tested in a Phase II clinical trial for PHN (NCT01195636) showing promising results (although statistical improvements in pain were not reported, a subgroup analysis showed that patients with a particular polymorphism in the Nav1.7 gene achieved statistical significance) [19]. The incidence of AEs was similar between funapide and placebo. However, application site-related AEs were more frequent in the placebo group than in the funapide treatment group. At the end of 2019, Flexion Therapeutics acquired funapide from Xenon Pharmaceuticals in order to develop a new candidate, FX301, which will combine funapide with a novel thermosensitive hydrogel for the treatment of postoperative pain [65]; this clinical trial started in 2021, but results are not yet available [66]. There are other VGSC blockers which showed efficacy for PHN in preclinical studies. One interesting example is tetrodotoxin, which is in Phase III for cancer-related pain [67] and showed promising results in a rat model of PHN [68].

#### 3.3.4. Cyclooxygenase-1 (COX-1) Inhibition

Cyclooxygenase inhibitors are effective and widely used for inflammatory pain treatment, but efficacy in NP is controversial [69]. Selective COX-2 inhibitors were effectively developed to solve some side effects (mainly gastrointestinal) but are associated with other security issues (such as cardiovascular side effects) [70]. Selective COX-1 inhibitors elicited less therapeutic interest, although some authors suggest that this therapeutic target should be reconsidered [71]. Interestingly, while the role of COX-2 in inflammatory pain is unquestionable, recent investigation suggests that COX-1 could be also implicated in inflammatory pain [72] and be even more important for NP [73]. Also, there is evidence of the fundamental role of COX-1 in animal models of pain with a neuropathic component [74,75]. TRK-700 is a COX-1 inhibitor developed by Toray Industries which was evaluated in a Phase II clinical trial for PHN [25]. TRK-700 showed efficacy in fibromyalgia and other models of NP [76], but human clinical results have not been published. It must be highlighted that unlike that reported for some analgesic drugs, TRK-700 did not show sedative effects.

#### 3.3.5. Adaptor-Associated Kinase 1 (AAK1) Inhibition

Preclinical studies have shown that the genetic depletion of AAK1 reduces pain in models of persistent pain but not acute pain without showing side effects or motor deficiencies [77]. In addition, AAK1 knockout mice were also resistant to the development of mechanical allodynia after spinal nerve ligation, confirming that AAK1 plays an important role in the development of NP [77]. It is suggested that the analgesic effect of AAK1 inhibitors is related with the reduced endocytosis of cell surface levels of μ2-containing GABA-A channels [77,78]. LX9211 is a potent AAK1 inhibitor developed by Lexicon Pharmaceuticals in collaboration with Bristol-Myers Squibb [79]. Results from two Phase I clinical trials were reported, and the most common AEs were mild: headache, dizziness and constipation. Nausea and vomiting were reported as moderate in severity, and they occurred more frequently compared to the placebo group [80]. A Phase II clinical trial is currently underway for DPN and another for NPH [78]. Recently, the company has announced that they are planning to continue with the development in Phase IIb and Phase III studies for DPN [81]. In addition, regarding preclinical evidence, it reduced thermal hyperalgesia in a nerve injury rat model and also reduced established mechanical allodynia in a rat model of DPN [77].

#### 3.3.6. Lanthionine Synthetase C-like Protein (LANCL) Activation

LANCL1 and LANCL2 are peptide-modifying enzymes that can promote cell protection and survival by reducing oxidative stress [82]. Studies at the neuronal level demonstrate that the LANCL1 transgene is neuroprotective by a mechanism related with glutathione [83,84,85]. LANCL1 was also identified as an immune marker of NP and may be a protective factor [86]. Meanwhile, the modulation of LANCL2 has been proposed as anti-inflammatory in some inflammatory conditions [87,88], with proved human safety [89]. Specifically, LANCL2 regulation reduced NP by regulating spinal neuroinflammation and nociceptive processing [90]. LANCL1 and LANCL2 activation could constitute a first-in-class target for analgesia. LAT8881 (AOD9604) is a synthetic C-terminal fragment of human growth hormone (GH) developed by Lateral Pharma that is in Phase II clinical trials for PHN (NCT03865953) [27]. This molecule is a particularly unique case, as no analgesic properties of a GH fragment were known. In fact, the neuroprotective and antioxidant effects of LAT8881 are not mediated by hormonal activity, they are attributed to its neuroprotective activity via a mechanism dependent on LANCL1 or LANCL2 [82]. LAT8881 showed efficacy in some preclinical models of inflammatory conditions such as arthritis [82,91], and it also showed good tolerability in both animal and human studies [92,93]. The results of the mentioned clinical trial were recently published showing disappointing results (no statistical differences versus placebo were found in any of the evaluated outcomes). The reported AEs included upper respiratory tract infection and headache, which were the same in both the placebo and LAT8881 treatment groups.

#### 3.3.7. N-Methyl-D-aspartate (NMDA) Receptor Antagonism

NMDA receptor antagonists have been suggested for the treatment of NP for years, as they participate in the transmission of pain signals [94,95]. In fact, the persistent stimulation of pain-involved receptors leads to the activation and positive regulation of synaptic NMDA receptors in the dorsal horn of the spinal cord, resulting in an amplification of pain signal transmission to the brain (central sensitization) [95]. Therefore, numerous clinical trials have been conducted with classic NMDA receptor antagonists that are already approved for other indications, such as memantine, dextromethorphan, amantadine, or ketamine [96,97]. Esketamine is the dextro form of ketamine. Ketamine is an NMDA receptor antagonist with a peculiarity: in addition to reducing pain by blocking the NMDA receptor, it can also activate inhibitory pain pathways and have anti-inflammatory effects [98]. Esketamine has a higher affinity for NMDA receptors and was recently approved as a nasal spray for the treatment of treatment-resistant depression [99]. Due to the higher affinity, the doses required to produce analgesia are lower. Ketamine has been used in the treatment of PHN [100,101], so a superior efficacy can be expected with esketamine. In this regard, there are some small clinical experiences for postoperative pain treatment where intranasal esketamine combined with intranasal midazolam was similar in effectiveness, satisfaction and safety compared with standard intravenous PCA with morphine. Intranasal esketamine spray did not report a difference in the number/severity of AEs compared to PCA with morphine. However, a high frequency of nystagmus was reported for the esketamine group and of dry mouth for the morphine group [31]. Also, there were case reports such as the case of a patient with potent PHN (9/10) in which treatment with esketamine along with trigeminal thermocoagulation produced a marked decrease in pain intensity to 2/10 that lasted for 2 months without adverse effects [102]. To generate evidence of its efficacy in a larger population, a Phase IV clinical trial of intranasal esketamine (NCT04664530) is underway including 48 participants with results expected by the end of 2023 [29].

#### 3.3.8. Mu Opioid Agonism

Mu opioids agonists are one of the oldest drugs ever (p.eg. morphine). Despite this, they are the most potent drugs for analgesia, and new structures with improved characteristics have been developed [103]. However, its efficacy in NP is less clear (second-line therapy in NeuPSIG guidelines) and even lesser for PHN (third-line therapy) [11]. Nevertheless, there are several new dosage forms of classical mu opioid agonists in clinical development for PHN. For example, a transdermal oxycodone patch was trialed in Phase IIa for PHN. The efficacy of oral oxycodone for the treatment of PHN was demonstrated [104], but transdermal administration presents some advantages (better adherence, less first-pass metabolism of the drug and less gastrointestinal side effects) [22]. Transdermal oxycodone was very safe with less systemic exposure (minor incidence of AEs for the oxycodone than for the control patch and any AE led to study discontinuation), but it did not produce analgesia for the broad PHN indication (only was effective in a subpopulation with high levels of paresthesia) [22]. Another case of novel dosage form was intravenous Patient-Controlled Analgesia (PCA) with hydromorphone, which was evaluated in a Phase IV clinical trial [23]. PCA can increase the improvement in pain intensity and patient satisfaction while similar rates of side effects were reported [105]. The clinical trial concluded that intravenous PCA hydromorphone provides a rapid onset of pain relief and can improve the current management of PHN [23]. The most frequent AEs in the hydromorphone group were nausea and dizziness, whilst the control group reported dizziness and somnolence. No patients exhibited respiratory depression in either group, and any AE led to study discontinuation. The last example is NZ-687, which is a sustained-release tramadol (bilayer formulation: 65% sustained release/35% immediate release) developed by Nippon Zoki Pharmaceutical Co. (Twotram^®^ tablets) [106] that was trialed in a Phase III study for PHN [21]. Tramadol is a particular case because it also acts as an SNRI, which makes this drug more interesting for NP treatment (second-line) [21]. NZ-687 was effective and well tolerated for PHN. Nausea, constipation, nasopharyngitis, somnolence, vomiting, and congestive heart failure were reported with no dose-dependent increase. Two patients discontinued tramadol treatment, while three discontinued in the placebo group [21].

#### 3.3.9. Nerve Growth Factor (NGF) Inhibition

It is well established, based on animal and human studies, that NGF is fundamental for nociception modulation [107]. A variety of NGF inhibitors, mostly monoclonal antibodies, have been developed and trialed for treating some musculoskeletal and non-musculoskeletal disorders (mainly osteoarthritis and low back pain) [108]. Despite several Phase II and III trials showed positive efficacy results using anti-NGF molecules (such as tanezumab), there is not any FDA-approved anti-NGF therapy due to safety concerns (rapid progression of osteoarthritis) [109]. However, the clinical usefulness of anti-NGF therapies for treating NP is less explored [108], and the one included in this systematic review (fulranumab; NCT00964990) is the first clinical trial inhibiting this target for treating PHN [30]. The results of this small Phase II clinical trial were disappointing (limited by the small sample size), and only some evidence of pain reduction was found at the highest dose of fulranumab (10 mg) [30]. Interestingly, fulranumab was well-tolerated in all doses (similar incidence of AEs between groups and no AE led to fulranumab discontinuation), so larger clinical studies are needed to assess efficacy in PHN [30].

#### 3.3.10. Other Targets

SR-419: Although the company states on its website and in press releases that the molecule is a first-in-class molecule, there is no known mechanism of action for this drug [28]. Additionally, there is no preclinical information of any kind on this molecule in different databases. With such opaque development as that being carried out by Shanghai SIMR Biotechnology, it is difficult to draw conclusions.

#### 3.3.11. Limitations

The main limitation of this systematic review is the lack of published information regarding some of the identified clinical trials. Some of them were completed several years ago, and the results were never published (e.g., TRK-700), which was possibly due to the absence of efficacy or safety concerns. In other cases, such as SR419, neither mechanism of action is reported. There are currently other ongoing clinical trials, and their results will be available soon (crisugabalin, LX9211 and esketamine). Another possible limitation may be that only one clinical trials registry (clinicaltrials.gov) and three different databases (Medline, Scopus and Web of Science) were explored, so it is possible that we missed some articles and clinical trials that were not included in these sources. The probability is however low, since clinicaltrials.gov is the largest available clinical trials registry, and all high-quality journals are indexed in the cited databases.

## 4. Conclusions

Since 2016, 15 different molecules have been evaluated in Phase II, III and IV clinical trials for PHN. Specifically, eight drugs were evaluated in Phase II (olodanrigan, funapide, TRK-700, LX9211, LAT8881, SR419, oxycodone and fulranumab), five were evaluated in Phase III (crisugabalin, mirogabalin, pregabalin, lidocaine and tramadol), and three were evaluated in Phase IV (pregabalin, esketamine and hydromorphone). Mirogabalin, modified-release pregabalin, lidocaine, tramadol and hydromorphone showed positive results while funapide, LAT8881, oxycodone patches and fulranumab lacked efficacy or had safety concerns (olodanrigan). Results are not published for crisugabalin, LX9211, SR419 and esketamine. Among them, we found four first-in-class targets for the treatment of PHN: AT2R antagonism, AAK1 inhibition, LANCL activation and NGF inhibition. These studies will result in novel drugs for a better pharmacological management of PHN.

## Figures and Tables

**Figure 1 ijms-24-12987-f001:**
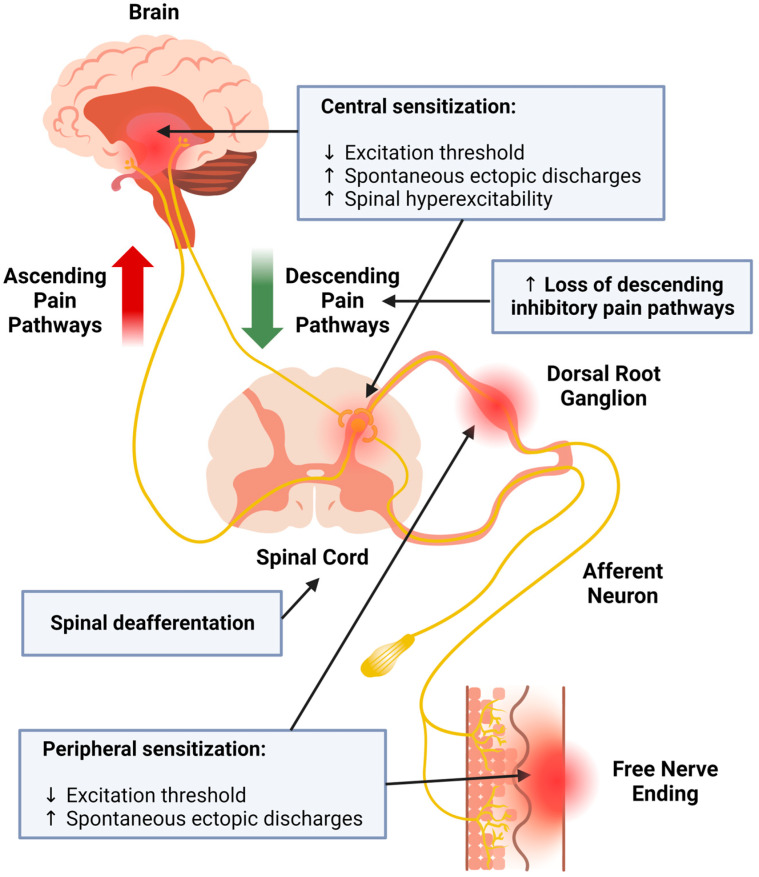
Summary of the sensory alterations experienced during postherpetic neuralgia.

**Figure 2 ijms-24-12987-f002:**
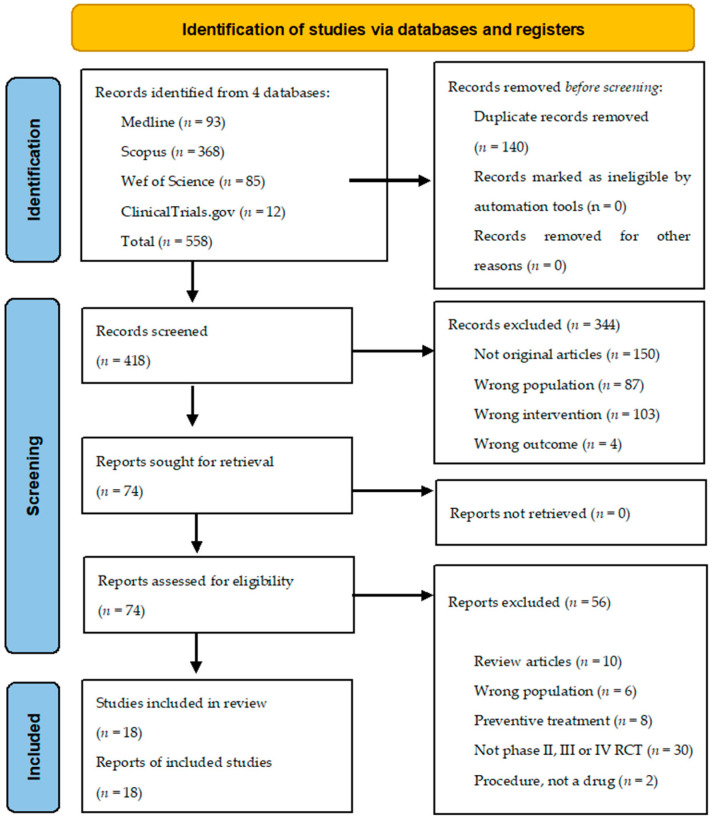
Study selection flow diagram.

**Table 1 ijms-24-12987-t001:** Main characteristics of included studies.

Main Pharmacological Target	Cite	Drug	Clinical Trial Code	Phase	Route and Dosage	Completion Date
* AT2R antagonist	[24]	Olodanrigan (EMA401)	NCT03094195	II	Oral: 25, 100, 300 mg BID	7 March 2019
VGCC α2δ subunit inhibitor	[13]	Crisugabalin (HSK16149)	NCT05140863	III	Oral: 4, 20 mg BID	5 January 2023
VGCC α2δ subunit inhibitor	[14]	Mirogabalin (DS-5565)	NCT02318719	III	Oral: 15, 20, 30 mg	25 May 2017
VGCC α2δ subunit inhibitor	[15]	Mirogabalin (DS-5565)	NCT02318719	III	Oral: 15, 20, 30 mg	25 May 2017
VGCC α2δ subunit inhibitor	[16]	Pregabalin (controlled release)	NCT01270828	III	Oral: 82.5 to 660 mg QD	November 2014
VGCC α2δ subunit inhibitor	[17]	Pregabalin	NCT01455428	IV	Oral: 300 mg/day BID	January 2014
VGCC α2δ subunit inhibitor	[18]	Pregabalin (YHD1119)	NCT02985216	III	Oral: 150, 300, 600 mg	2 May 2018
VGSC blocker (Nav1.6/1.7)	[19]	Funapide (TV-45070; XEN402; XPF-002)	NCT01195636	IIa	Topical: ointment 8% BID	March 2011
VGSCs blocker	[20]	Lidocaine	ChiCTR 1800017762	III	Intravenous: 4 mg/kg QD	June 2021
COX-1 inhibitor	[25]	TRK-700	NCT02701374	II	Oral: not specified	July 2017
* AAK1 inhibitor	[26]	LX9211	NCT04662281	II	Oral: not specified QD	21 December 2022
* LANCL ligand	[27]	LAT8881 (AOD9604)	NCT03865953	II	Oral: 30 mg BID	3 May 2020
Unknown	[28]	SR419	NCT05357677	II	Oral: 30 mg TID	18 January 2023
NMDAR antagonist	[29]	Esketamine	NCT04664530	IV	Intranasal: 5–35 mg TID	1 December 2023
Mu opioid agonist and SNRI	[21]	Sustained-release tramadol (NZ-687)	JapicCTI-163341	III	Oral: 100–400 mg/day BID	7 December 2022
Mu opioid agonist	[22]	Oxycodone	ACTRN 12615000013561	IIa	Transdermal: 23.4 mg in 3 d	18 April 2017
Mu opioid agonist	[23]	Hydromorphone (PCA)	ChiCTR 1800019880	IV	Intravenous PCA: 1 mg/mL	30 June 2020
* mAB anti-NGF	[30]	Fulranumab (JNJ-42160443)	NCT00964990	II	Subcutaneous: 1, 3, or 10 mg every 28 d	2 May 2016

* Considered first in class drugs for the treatment of pain. AT2R: Angiotensin II Type 2 Receptor; VGCC: Voltage-Gated Calcium Channel; VGSC: Voltage-Gated Sodium Channel; COX-1: Cyclooxygenase-1; AAK1: Adaptor-Associated Kinase 1; LANCL: Lanthionine Synthetase Component C-Like Protein; NMDAR: N-Methyl-D-Aspartate Receptor; SNRI: Serotonin and Norepinephrine Reuptake Inhibitor; mAB: monoclonal antibody; NGF: Nerve Growth Factor; PCA: patient-controlled analgesia; QD: once a day; BID: twice a day; TID: three times a day.

**Table 2 ijms-24-12987-t002:** Information about primary outcomes regarding efficacy in the included studies.

Drug	Main Result (Primary Outcome)	Other Information (Primary Outcome)
Olodanrigan (EMA401) [24]	NRS TD: 25 mg: −0.5 (95% CI: −1.6, 0.7; *p* = 0.408) 100 mg −0.6 (95% CI: −1.6, 0.5; *p* = 0.308)	Reduction in all doses for LS means (change from baseline in the 24 h average weekly mean NRS pain score over 12 weeks)
Crisugabalin (HSK16149) [13]	Clinical results for PHN not published Only preclinical evidence available	Preclinical evidence compared to pregabalin: Greater efficacy Fewer central AEs
Mirogabalin (DS-5565) [14,15]	LS in ADPS mean versus placebo was: 15 mg/day: −0.41 (*p* = 0.0170) 20 mg/day: −0.47 (*p* = 0.0058) 30 mg/day: −0.77 (*p* < 0.0001)	Responders rate (≥30% ADPS): 35.0% for placebo 45.4% for 15 mg/day 45.1% for 20 mg/day 49.7% for 30 mg/day
Pregabalin (CR) [16]	LTR criteria (*p* < 0.0001): 29 patients (13.9%) with pregabalin CR 63 patients (30.7%) with placebo	Time to loss of therapeutic response (LTR): <30% pain response or Discontinuation (AE or lack of efficacy)
Pregabalin [17]	Reduction in mean pain score: Greater with pregabalin vs. placebo LS TD −0.71 (95% CI: −1.08, −0.34; *p* = 0.0002)	Proportion of patients with ≥30% mean pain reduction (*p* = 0.0007): Pregabalin: 52.3% Placebo: 30.6%
Pregabalin (YHD1119) [18]	LS mean in DPRS score at the end: SR pregabalin: 3.01 IR pregabalin: 3.06 TD of 0.06; 95% CI: −0.31 to 0.42	SR pregabalin was not inferior to IR pregabalin in reducing pain intensity (*p* non-inferiority < 0.0001)
Funapide (TV-45070; XEN402; XPF-002) [19]	LS mean change in mean daily pain score: Placebo: −0.97 TV-45070: −0.94 (*p* = 0.8885)	No TD in the primary efficacy analysis Post hoc subgroup analysis: significant improvements in R1150W polymorphism heterozygous carriers versus wild-type counterparts and placebo
Lidocaine [20]	Pain reduction after infusion and TD superior vs. placebo (*p* < 0.0001)	NRS pain scores and 24 h breakthrough pain numbers were lower in the lidocaine group (*p* = 0.001)
TRK-700 [25]	No results published. Recruitment completed on 2017-07	Primary outcome: change in average NRS from baseline to week 8
LX9211 [26]	No results published Recruitment completed on 2022-12-21	Primary outcome: change from baseline (week 2) to week 6 in ADPS (11-point NRS)
LAT8881 (AOD9604) [27]	Primary outcome: mean change in pain intensity scores (NPRS) during 4 weeks: No TD was found (*p* = 0.67) LAT8881 group: reduction of −0.87 Placebo group reduction of −0.74	30% responder rate was not significant: 20/50 in treatment 19/50 in placebo and 50% responder rate was not significant: 6/50 in treatment 9/50 in placebo
SR419 [28]	No results published Recruitment completed on 2023-03-19	Primary outcome: proportion of subjects who rate their pain as “much improved” or “very much improved” (PGIC)
Esketamine [29]	No results published yet Not finished (now recruiting) Estimated completion date: 2023-12-01	NRS will be measured at baseline and once for the period of drug administration
Sustained-release tramadol (NZ-687) [21]	Proportion of patients with an inadequate analgesic effect (double-blind period): Tramadol: 16.9% (95% CI 9.5%–26.7%) Placebo: 39.8% (95% CI 29.5%–50.8%)	Cumulative retention rate: Greater in tramadol group (*p* = 0.0005) ≥20% greater than those in the placebo group
Oxycodone [22]	No improvement in NPRS (oxycodone patch vs. vehicle patch)	Reduced average pain scores in high level paresthesia subpopulation (post hoc) (*p* < 0.05)
Hydromorphone (PCA) [23]	NRS 1, 4 and 12 weeks after treatment: Control: 4.5 ± 1.4, 3.5 ± 1.3, and 3.0 ± 1.0, Drug: 3.3 ± 1.1, 2.8 ± 0.6, and 2.1 ± 0.5	NRS scores in between groups had a significant difference in 1, 4 and 12 weeks after treatment (all *p* < 0.001)
Fulranumab (JNJ-42160443) [30]	No significant TD or dose–response was observed in responders (≥30% improvement): 38.5% for 1 mg (5/13) 15.4% for 3 mg (2/13) 21.1% for 10 mg (4/19)	For ≥50% improvement difference vs. placebo 15.4% for 1 mg 15.4% for 3 mg 15.8% for 10 mg

NRS: Numerical Rating Score; TD: Treatment Difference; LS: Least Squares; PHN: Postherpetic Neuralgia; AE: Adverse Event; ADPS: Average Daily Pain Score; LTR: Time to Loss of Therapeutic Response; CR: Controlled Release; CI: Confidence Interval; DPRS: Daily Pain Rating Score; SR: Sustained Release; IR: Immediate Release; NPRS: Numerical Pain Rating Score; PGIC: Patients’ Global Impression of Change.

**Table 3 ijms-24-12987-t003:** Main information regarding drug safety in the included studies.

Drug	Overall Safety Information (vs. Placebo)	Most Frequent AEs
Olodanrigan (EMA401) [24]	Patients that experienced ≥1 AE: 65.1% in placebo 58.1% in EMA401 25 mg 62.8% in EMA401 100 mg Discontinuation due to an AE: 2.3% in placebo 7.0% in EMA401 25 mg 7.0% in EMA401 100 mg	Diarrhea: 7.0% in placebo 7.0% in EMA401 25 mg 4.7% in EMA401 100 mg Nasopharyngitis: 9.3% in placebo 7.0% in EMA401 25 mg 4.7% in EMA401 100 mg Muscle spasms: 4.7% in EMA401 25 mg Increased levels of amylase, lipase, triglycerides, and creatinine in blood only in EMA401 groups (low frequency 0–7%)
Crisugabalin (HSK16149) [13]	No publications available for PHN Safety information extracted from a Phase II/III DPN trial	Dizziness Somnolence (All transient and mild, no treatment needed)
Mirogabalin (DS-5565) [14,15]	Total discontinuation: 12.5% in placebo 3.9% in DS-5565 15 mg/day 7.8% in DS-5565 20 mg/day 2.6% in DS-5565 30 mg/day Discontinuation due to an AE: 4.0% in placebo 5.3% in DS-5565 15 mg/day 10.5% in DS-5565 20 mg/day 7.7% in DS-5565 30 mg/day	Dizziness Somnolence Weight increase Edema (All mild/moderate, more frequent for DS-5565 groups in a dose-dependent manner and resolved without treatment)
Pregabalin (CR) [16]	Patients that experienced ≥1 AE: 30.7% in placebo 38.5% in pregabalin (CR) Discontinuation due to an AE: 2.9% in placebo 1.4% in pregabalin (CR)	Edema: 0.5% in placebo 3.8% in pregabalin (CR) Weight gain: 1.0% in placebo 3.8% in pregabalin (CR) Dizziness: 0.5% in placebo 3.4% in pregabalin (CR) Nausea: 0% in placebo 3.4% in pregabalin (CR)
Pregabalin [17]	Patients that experienced ≥1 AE: 44.0% in placebo 64.0% in pregabalin Premature discontinuation due to an AE: 1.8% in placebo (pain) 5.4% in pregabalin (4 dizziness, 1 RTI and 1 cerebral ischemia)	Nasopharyngitis: 8.3% in placebo Somnolence and pruritus: 4.6% in placebo Dizziness: 24.3% in pregabalin Peripheral edema: 6.3% in pregabalin
Pregabalin (YHD1119) [18]	Patients that experienced ≥1 AE: 50.0% in IR pregabalin 52.7% in SR pregabalin All mild/moderate	Dizziness: 17.7% in IR pregabalin 28.8% in SR pregabalin Somnolence: 5.9% in IR pregabalin 8.7% in SR pregabalin
Funapide (TV-45070; XEN402; XPF-002) [19]	Patients that experienced ≥1 AE (non-DR): 50.8% in placebo 53.2% in TV-45070 Patients that experienced ≥1 AE (DR): 17.7% in placebo 30.2% in TV-45070 Discontinuation due to an AE: 5 in placebo (cutaneous) 3 in TV-45070 (2 cutaneous; 1 CAD non DR)	General disorders and application site-related (e.g., pain and pruritus) 27% in placebo 19.4% in TV-45070 Nervous system disorders (dizziness and headache): 4.8% in placebo 6.5% in TV-45070 Infections and infestation: 17.5% in placebo 11.3% in TV-45070
Lidocaine [20]	No differences in AEs vs. placebo were reported All 32 patients in the lidocaine group received ≥5 lidocaine infusions AEs were registered during and 30 min after infusions	Somnolence: 25% in lidocaine Dry mouth: 18.8% in lidocaine Peripheral numbness: 9.4% in lidocaine Dizziness: 6.3% in lidocaine Tinnitus: 3.1% in lidocaine Chest tightness: 3.1% in lidocaine
TRK-700 [25]	No publications available	Without CNS alterations (All in rat models of DPN and fibromyalgia)
LX9211 [26]	Data not published Safety information extracted from two Phase I studies Patients that experienced ≥1 AE: 25.0% in placebo 31.9% in LX9211	Headache: 15% in LX9211 Dizziness: 10% in LX9211 Bowel movements: 10% in LX9211
LAT8881 (AOD9604) [27]	Patients that experienced ≥1 AE: 11.8% in placebo 9.8% in AOD9604	Upper RTI (nasopharyngitis) Headache (In both groups)
SR419 [28]	Data not published	Data not published
Esketamine [29]	Data not published No differences between esketamine and morphine (study for postoperative pain [31])	A tendency to a higher frequency of nystagmus in esketamine A tendency to a higher frequency of dry mouth in morphine
SR tramadol (NZ-687) [21]	Total AEs: 39.8% in placebo 33.7% in tramadol Premature discontinuation due to an AE: 3 in placebo (sleep disorder, akathisia, and spondylolisthesis) 2 in tramadol (somnolence/nausea and congestive heart failure)	Nausea: 7.2% in NZ-687 Vomiting: 2.4% in NZ-687 Constipation: 6.0% in NZ-687 Nasopharyngitis: 3.6% in NZ-687 Somnolence: 2.4% in NZ-687
Oxycodone [22]	Total AEs: 44.4% in control 37.0% in oxycodone Patients that experienced ≥1 AE: 51.9% in control 40.7% in oxycodone	General disorders and administration site conditions: 37.0% in control 18.5% in oxycodone Respiratory disorders: 7.4% in oxycodone (not in control) (All mild/moderate)
Hydromorphone (PCA) [23]	Incidence 1 week after treatment: 8.2% in control 20.8% in hydromorphone (PCA) Incidence 4 weeks after treatment: 6.2% in control 3.1% in hydromorphone (PCA) Incidence 11 weeks after treatment: 6.2% in control 1.0% in hydromorphone (PCA)	Dizziness and somnolence Nausea and vomiting Sweating Constipation Urinary retention Drowsiness (All transient and improved after treatment)
Fulranumab (JNJ-42160443) [30]	Patients that experienced ≥1 AE: 80% in placebo 62% in JNJ-42160443 1 mg 62% in JNJ-42160443 3 mg 79% in JNJ-42160443 10 mg	Osteoarthritis: >10% difference between JNJ-42160443 10 mg and placebo groups Headache, arthralgia, back pain and motor-related (also in placebo, no TD) (All mild/moderate)

AE: Adverse Event; CAD: Coronary Artery Disease; CNS: Central Nervous System; CR: Controlled-Release; DPN: Diabetic Peripheral Neuropathy; DR: Drug-Related; IR: Immediate Release; PCA: Patient-Controlled Analgesia; PHN: Postherpetic Neuralgia; RTI: Respiratory Tract Infection; SR: Sustained Release; TD: Treatment Difference.

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
