# Peer review of "Investigational Drugs for the Treatment of Postherpetic Neuralgia: Systematic Review of Randomized Controlled Trials"

_ijms, 2023, doi:10.3390/ijms241612987_

Round 1
Reviewer 1 Report
The manuscript by Huerta et al. nicely reviews the 18 articles that conducted clinical trials evaluating drugs for the treatment of postherpetic neuralgia (PHN). The authors introduce the pharmacological or biological effects of the drugs classified based on 10 different mechanisms. I feel this may be welcomed by the journal's audience, but I have several comments as below.
Specific comments
1. The results section should contain the “results” but not the method. They had better move the sentences of the results in the original manuscript into “Materials and Methods” section.
2. The discussion seems to be “Results and Discussion” in this original manuscript.
3. It is hard to read the table 2. The authors should reedit the Table 2 specifically the part of “Main results”. The style of the description should be unified or bulleted as much as possible (e.g., the assessment; dose; statistical significance, etc.).
Author Response
We thank the reviewer for the positive comments and their helpful suggestions, which have helped us to improve the manuscript. All the specific comments have been considered and addressed in the revised manuscript as in the document attached.

Reviewer 2 Report
This manuscript is a systematic review that summarizes the results of new drugs evaluated in phases II, III, and IV clinical trials conducted between 2016 and 2023. The review reports the molecules and pharmacological targets that may be available in the future therapeutic arena. The authors have searched 3 electronic databases (Medline, Web of Science, Scopus) and the register of ClinicalTrials.gov. They found 18 clinical trials in them 14 molecules with pharmacological actions on 9 different molecular targets were identified and presented. This is a valuable review that provides the status of the pharmacological treatment of postherpetic neuralgia (PHN). Since no optimal treatment is available, these agents might be promising agents for future treatment. The authors have followed the guidelines for systematic review and registered their protocol and reported the process as of recommended with PRISMA and used Rayyan.
Please consider the following points in the revision:
The study selection flow diagram shows 342 excluded items. Please provide the reason (s).
Please add why 3 databases are selected. Based on the literature (please see the paper here: https://systematicreviewsjournal.biomedcentral.com/articles/10.1186/s13643-017-0644-y), it is recommended to have at least 4 databases. Please elaborate.
Please indicate why the European clinical trial register (https://www.clinicaltrialsregister.eu/) has not been included and only FDA clinical trial register has been searched.
The graphical abstract is valuable in the visualization of the compounds and their effects. Perhaps, the authors can consider a short section or a schematic related to the PHN mechanism (those that are known or proposed) so that understanding the mechanism of action of drugs and pharmacological targets will be easy to understand.
Related to side effects and some serious adverse effects, can the authors add an insight? they have summarized in the table, however, in the discussion, safety, and efficacy can be discussed to draw attention that efficacy and safety are both determinants in the overall outcome for a compound to be therapeutically used.
Please add the limitations of your systematic review.
Please add the statement of “the review question” and justification of this particular review as to what exact reason the authors aimed at conducting this review. These are the points that are proposed that the authors have already registered in PROSPERO, but worth adding in the text for clarification for the readers.
Please proof read the text for few editorial errors.
Author Response
We thank the reviewer for their thoughtful suggestions. A point-by-point response to the comments is given below and all issues have been addressed in the revised manuscript as explained below. We expect to have addressed them properly and you find the manuscript now suitable for publication. Clearly, because of these changes, the quality of the manuscript has improved a lot.

Round 2
Reviewer 1 Report
Authors appropriately responded and answered to my comments. The same comments as the Table 2 will be applied to the Table 3. After revise the Table 3, I think the revised manuscript will be suitable for publication in the International Journal of Molecular Sciences.
Author Response
We thank the reviewer for the positive comments and their helpful suggestions, which have helped us to improve the manuscript. The last suggestion has been considered and addressed in the revised manuscript as explained in the attached document.
